# Protein or ribonucleoprotein-mediated blocking of recombinase polymerase amplification enables the discrimination of nucleotide and epigenetic differences between cell populations

Toshitsugu Fujita [1✉], Shoko Nagata[1] & Hodaka Fujii [1✉]

Isothermal DNA amplification, such as recombinase polymerase amplification (RPA), is well suited for point-of-care testing (POCT) as it does not require lengthy thermal cycling. By exploiting DNA amplification at low temperatures that do not denature heat-sensitive molecules such as proteins, we have developed a blocking RPA method to detect gene mutations and examine the epigenetic status of DNA. We found that both nucleic acid blockers and nuclease-dead clustered regularly interspaced short palindromic repeats (CRISPR) ribonucleoproteins suppress RPA reactions by blocking elongation by DNA polymerases in a sequence-specific manner. By examining these suppression events, we are able to discriminate single-nucleotide mutations in cancer cells and evaluate genome-editing events. Methyl-CpG binding proteins similarly inhibit elongation by DNA polymerases on CpG-methylated template DNA in our RPA reactions, allowing for the detection of methylated CpG islands. Thus, the use of heat-sensitive molecules such as proteins and ribonucleoprotein complexes as blockers in low-temperature isothermal DNA amplification reactions markedly expands the utility and application of these methods.

[1] Department of Biochemistry and Genome Biology, Hirosaki University Graduate School of Medicine, Aomori, Japan. ✉email: toshitsugu.fujita@hirosaki-u.ac.jp; hodaka@hirosaki-u.ac.jp

Methods for the amplification of target DNA sequences, such as the polymerase chain reaction (PCR), are frequently used in a variety of fields, including molecular and medical biology. However, PCR requires specialized equipment such as a thermal cycler, making it unsuitable for certain point-of-care testing (POCT) applications, such as the detection of pathogenic microbial DNA to aid the diagnosis of infectious diseases in remote, rural areas, where access to a thermal cycler is limited. Conversely, isothermal DNA amplification methods such as loop-mediated isothermal amplification (LAMP)[1] and recombinase polymerase amplification (RPA)[2] do not require such dedicated equipment, because the target DNA is amplified at a constant temperature (approximately 60 °C and 37 °C, respectively). In addition, RPA completes DNA amplification within 20 min in most cases[3], which is advantageous over PCR. Despite these advantages, the application of RPA remains limited.

Nucleotide differences can be detected by PCR using sequence-specific blockers, such as locked nucleic acids (LNA) and peptide nucleic acids (PNA)[4]. An LNA/PNA complementary to a target wild-type (WT) DNA sequence hybridizes with the sequence and blocks recruitment of a forward/reverse primer or DNA extension across the sequence by DNA polymerases. If a target DNA possesses a nucleotide difference (e.g., a mutation), which results in a mismatch(es) with an LNA/PNA, such blockage will not occur (i.e., DNA will be amplified). These sequence-specific blockers can also be used in blocking RPA; a modified RPA reaction that includes an additional heating/cooling step for PNA blocker–target DNA hybridization, which blocks recruitment of a primer[5]. However, because a single-nucleotide mismatch between a blocker and target DNA sequence would be easily tolerated due to the additional heating/cooling step, it may be difficult to clearly discriminate a single-nucleotide difference in a target DNA sequence by this method. Moreover, the advantages of using isothermal DNA amplification over PCR are lost. A clamp primer consisting of an oligodeoxyribonucleotide (ODN) with a specific 3'-modification can similarly be used to block annealing of a primer, enabling the discrimination of single-nucleotide differences by RPA[6]. However, engineering overlapping annealing sites for clamp, forward, and reverse primers reduce flexibility in the design of these primers. Therefore, while blocking RPA has the potential for the detection of nucleotide differences between samples, additional improvements are required. The use of heat-sensitive molecules as blockers may markedly improve the utility and application of low-temperature isothermal DNA amplification reactions over PCR and high-temperature isothermal methods.

In this study, to expand the utility and flexibility of low-temperature isothermal DNA amplification, we examine oligoribonucleotides (ORN, approximately 20 nt RNA) as sequence-specific blockers to block DNA extension by DNA polymerases for the detection of nucleotide differences by blocking RPA (Fig. 1a–c). We previously showed that ORNs can block DNA extension by DNA polymerases in PCR reactions[7–9]. Taking advantage of the low isothermal temperature of RPA reactions, we identify heat-sensitive proteins and ribonucleoprotein (RNP) complexes, namely, nuclease-dead forms of clustered regularly interspaced short palindromic repeats (CRISPR) RNP complexes[10,11], as sequence-specific blockers. In this context, the nuclease-dead form of *Streptococcus pyogenes* Cas9 (dCas9) and guide RNA (gRNA) are employed. We also create a blocking RPA protocol using heat-sensitive proteins as a tool for the discrimination of epigenetic marks on target DNA. Thus, we show that RPA is compatible with a variety of heat-sensitive protein and RNP blockers, expanding the range of biological and medical applications, such as cancer diagnosis and epigenetic analysis, that can be conducted without a thermal cycler.

## Results

### Discrimination of a single-nucleotide mutation by blocking RPA using an ORN

We first examined ORNs as sequence-specific nucleotide blockers to RPA of the human *KRAS* gene (Fig. 1a and b and Fig. 2a and b). RPA in the absence of ORNs amplified a sequence across human *KRAS* codon 13 (Gly13 or G13) and codon 12 (Gly12 or G12) from genomic DNA (gDNA) extracted from 293T cells, which have two WT alleles of *KRAS* (Fig. 2c). However, in the presence of an ORN that targets the Gly13 sequence and aligns perfectly with the WT allele (ORN_KRAS, 4 μM)[9], *KRAS* amplification was completely suppressed (Fig. 2b and c and Supplementary Fig. 1). Neither irrelevant ORNs nor an ORN possessing a single-nucleotide mutation in ORN_KRAS (ORN_KRAS _mut) suppressed *KRAS* amplification (Fig. 2b–d). These results show sequence-specific blocking of RPA by ORNs. We next used gDNA from the cell line HCT116, which has a heterozygous Gly13Asp (G13D (GGC > GAC)) *KRAS* mutation (Fig. 2a and b), to examine the discrimination of a single-nucleotide mutation by blocking RPA using an ORN (Fig. 1b and c). RPA blocked by ORN_KRAS specifically amplified the G13D *KRAS* allele (Fig. 2e and f), suggesting that the blocking of RPA using an ORN can be used to detect the presence of a single-nucleotide mutation. We refer to this method of blocking RPA as "ORN-interference RPA (ORNi-RPA)". Unlike PNAs[5], an ORN is easy to design[9] and cost-effective to synthesize. In addition, because it is recruited to its target sequence by the recombinase and the single-strand binding protein (SSB) present in RPA reactions, the additional heating/cooling step for hybridization is not necessary. Therefore, the ability to use ORNs as a blocking agent would improve the utility and flexibility of blocking RPA.

To further examine the feasibility of ORNi-RPA, we next used a commercially available human gDNA that heterozygously possesses another well-known Gly12Asp (G12D (GGT > GAT)) mutation in the *KRAS* gene (Fig. 2a). As shown in Fig. 2g and h, ORNi-RPA with ORN_KRAS#2 (Fig. 2b) specifically amplified the G12D *KRAS* allele, whereas that using ORN_KRAS suppressed amplification of both the WT and G12D *KRAS* alleles. When we tested another ORN, called ORN_KRAS#3, ORNi-RPA amplified the G12D, but not the G13D, *KRAS* allele (Supplementary Fig. 2a–d). Thus, these results demonstrate that ORNi-RPA can discriminate a single-nucleotide difference when an appropriate ORN is designed. ORNs tolerated single-nucleotide mismatches in some cases (Supplementary Fig. 2e). In addition, because the effective concentrations of ORNs differed (Supplementary Figs. 1 and 2b), titration of an ORN would be beneficial to optimize the assay systems. Considering these findings, we propose a potential step-by-step procedure for ORNi-RPA to detect a single-nucleotide difference (Supplementary Fig. 2f), which would be useful to target a new sequence.

### Discrimination of single-nucleotide mutations by blocking RPA using RNP complexes

The successful suppression of RPA with an ORN prompted us to explore other molecules as inhibitors of RPA. One of the advantages of RPA over PCR and high-temperature isothermal DNA amplification methods, such as LAMP, is that the reaction mixture is incubated at 37 °C and heat-sensitive molecules such as proteins can be added to the reaction mixture. To investigate the potential inhibition of RPA by heat-sensitive molecules, we first tested nuclease-dead forms of CRISPR RNP complexes[10,11] as sequence-specific blockers (Fig. 1b and Supplementary Fig. 3a). We utilized *S. pyogenes* dCas9 and gRNA consisting of CRISPR RNA (crRNA) and trans-activating crRNA (tracrRNA) sequences that are functional at 37 °C[12]. As shown in Fig. 3a and b, *KRAS* amplification was specifically suppressed by

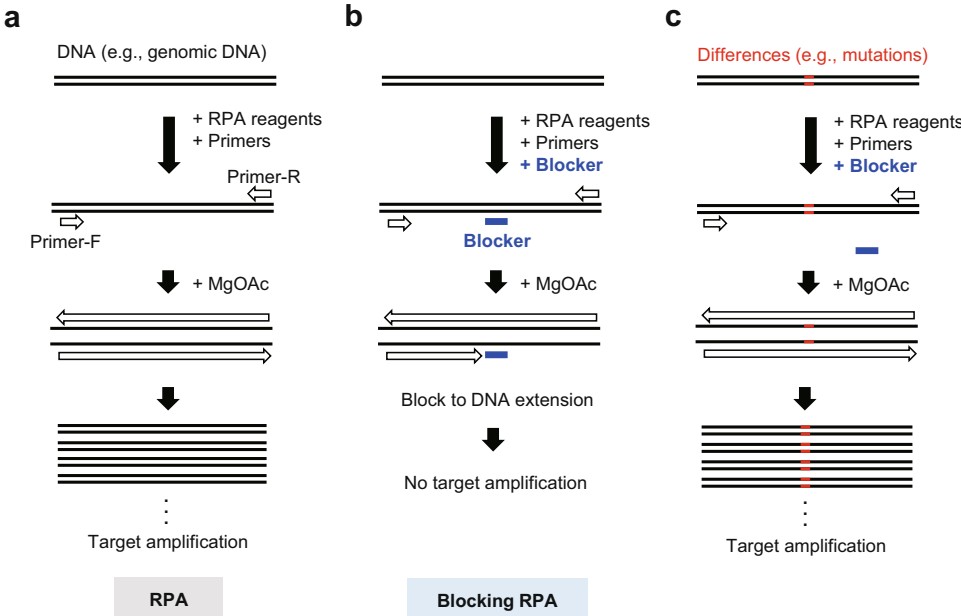

**Fig. 1 Schematic diagram showing blocking RPA and its application. a** During RPA, DNA polymerases extend DNA from primers, recruited to their complementary sequences by recombinases, in a magnesium acetate (MgOAc)-dependent manner. **b** In the presence of a sequence-specific blocker (e.g., ORN), DNA extension by DNA polymerases is inhibited, resulting in the suppression of DNA amplification across the target sequence. **c** If the target sequence is mutated, it cannot be recognized by the blocking agent, and DNA amplification proceeds uninhibited (allowing for the detection of a DNA mutation).

CRISPR RNP complexes targeting nucleotides around the Gly13-coding sequence of WT human *KRAS* (gRNA_KRAS and gRNA_KRAS#2), but neither dCas9 nor gRNA alone suppressed amplification (Supplementary Fig. 3b and c). These data show that CRISPR RNP complexes effectively block amplification by DNA polymerases in a sequence-specific manner in RPA reactions. Hereafter, we refer to this in vitro blocking method as "CRISPR interference (CRISPRi)-RPA" to distinguish it from conventional in vivo applications of CRISPRi[10,11].

We next applied CRISPRi-RPA to the discrimination of single-nucleotide mutations. CRISPRi-RPA with gRNA_KRAS#2 amplified the G13D but not WT *KRAS* sequence from HCT116 gDNA (Fig. 3c and d), suggesting that gRNA_KRAS#2 discriminates the single-nucleotide mutation. However, CRISPRi-RPA with gRNA_KRAS displayed blocked amplification of both alleles, as the single-nucleotide mismatch between the G13D *KRAS* sequence and the gRNA sequence was tolerated, and the gRNA was able to bind efficiently (Fig. 3a and c). The single-nucleotide mutation responsible for the G13D *KRAS* allele falls within the proto-spacer adjacent motif (PAM) sequence of gRNA_KRAS#2 and within the region of gRNA_KRAS that targets *KRAS* for binding (Fig. 3a). This suggests that for CRISPRi-RPA, the PAM sequence is critical for the discrimination of single-nucleotide mutations (see Fig. 3g). Interestingly, gRNA_KRAS_mut, a gRNA mutated to intentionally form one or two nucleotide mismatch(es) between the target DNA and gRNA sequences (Fig. 3a), successfully discriminated the single-nucleotide substitution of *KRAS* (Fig. 3e and f). In this regard, the two nucleotide mismatches were not tolerated, and the gRNA was unable to bind the G13D *KRAS* sequence efficiently (Fig. 3g). To further examine the feasibility of CRISPRi-RPA, we next used human gDNA that possesses a heterozygous G12D *KRAS* mutation. As shown in Supplementary Fig. 4a–c, CRISPRi-RPA specifically amplified the G12D *KRAS* allele when the single-nucleotide mutation responsible for the G12D *KRAS* allele fell within the PAM sequence of a gRNA (gRNA_KRAS#3). In addition,

CRISPRi-RPA with gRNA_KRAS_mut also amplified the G12D *KRAS* allele (Supplementary Fig. 4a, d, and e). On the other hand, we have previously shown that a CRISPR complex consisting of Cas9 and gRNA_p16_Gx5#2, a gRNA targeting the human *CDKN2A (p16)* gene (Supplementary Fig. 5a), discriminates a single-nucleotide insertion/deletion at the target binding step, allowing for allele-specific genome editing[13]. Taking advantage of this property of the gRNA, we successfully discriminated a single-nucleotide insertion/deletion in the *CDKN2A (p16)* sequence by CRISPRi-RPA (Supplementary Fig. 5b–d). Taken together, these data show that CRISPRi-RPA is useful as a tool to detect single-nucleotide mutations.

Next, we examined the sensitivity of CRISPRi-RPA to discriminate nucleotide differences. As shown in Supplementary Fig. 6a–d, CRISPRi-RPA with gRNA_KRAS#2 detected the mutated G13D *KRAS* so long as >0.5% of copies of the *KRAS* gene had the mutation. In addition, we found that 40 ng of dCas9, which was the dose used throughout this study, was sufficient to suppress amplification of a target sequence (e.g., WT *KRAS*) from less than 0.4 μg of human gDNA (Supplementary Fig. 6a, lane 4 and 7, and 6b) and only 4 ng of human gDNA was required for CRISPRi-RPA to amplify the desired sequence (e.g., mutated G13D *KRAS*) (Supplementary Fig. 6c, lane 4 and 5, and 6d). Taking these properties into consideration, we conclude that CRISPRi-RPA can be applied to the detection of pathological cells possessing single-nucleotide mutations, such as cancer cells, in environments where WT cells outnumber these mutants.

**Evaluation of genome editing events by CRISPRi-RPA.** We further applied CRISPRi-RPA to the evaluation of genome editing events, replacing T7 endonuclease I or Surveyor assays[14–16]. Genome editing events mediated by *S. pyogenes* Cas9 and gRNA introduce mutations in the gRNA target site. Therefore, we wanted to examine whether CRISPRi-RPA using the same gRNA would be able to amplify only mutated sequences (Supplementary

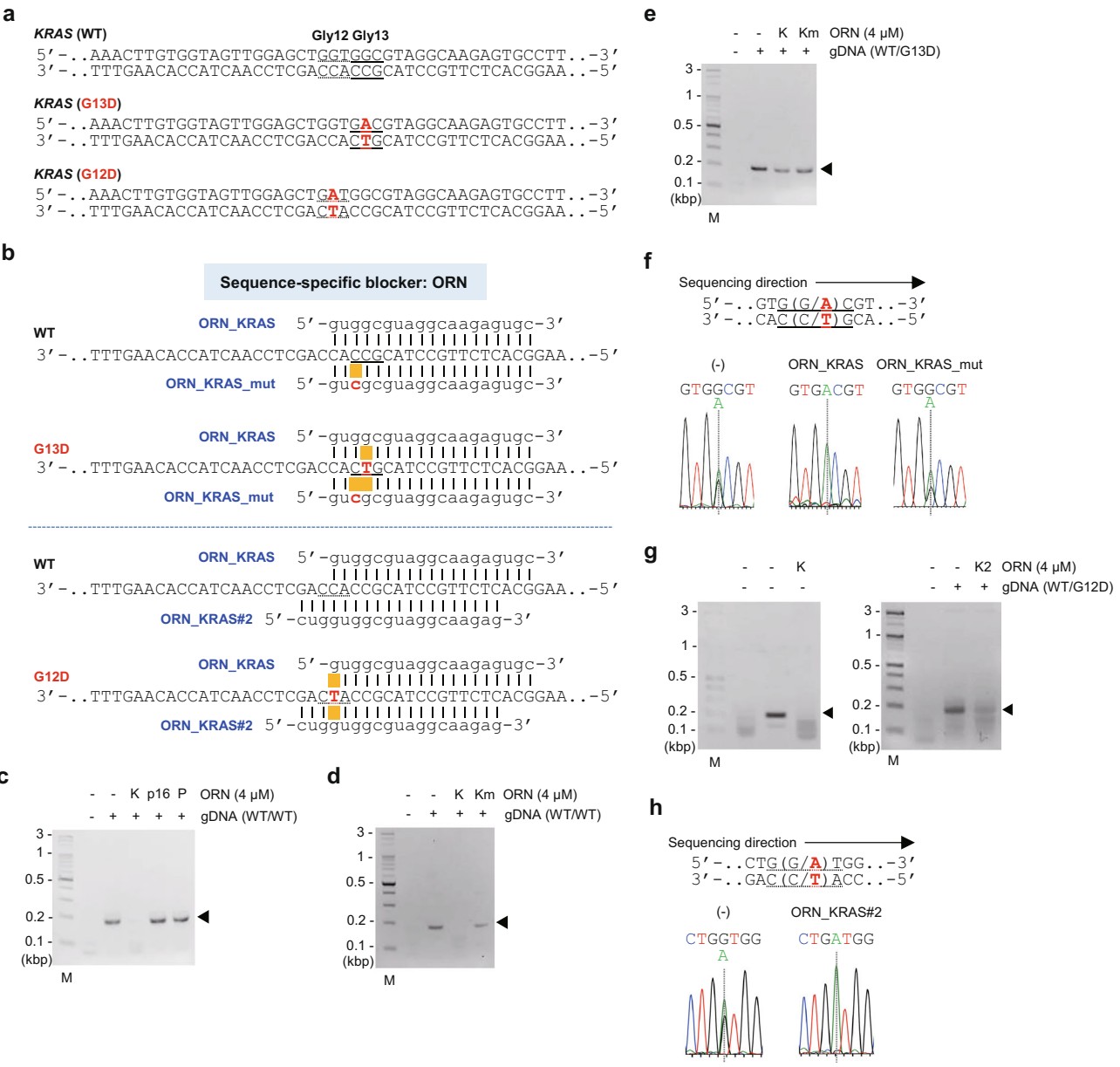

**Fig. 2 Blocking RPA using ORNs as blocking agents. a** The WT and mutated sequences of the human *KRAS* gene. **b** ORNs examined as sequence-specific blocking agents and their complementary sequence in the human *KRAS* gene. **c**, **d** Results of RPA blocked by an ORN (ORNi-RPA) using template gDNA extracted from 293T cells, which are WT for *KRAS*. ORNs targeting human *KRAS* (ORN_KRAS; K), human *CDKN2A (p16)* (ORN_p16; p16), chicken *Pax5* (ORN_cPax5_Ex1B; P), and an ORN that differs from ORN_KRAS by one nucleotide (ORN_KRAS_mut; Km) were used. Amplified *KRAS* is indicated by an arrowhead. M, molecular weight marker. **e** Results of ORNi-RPA of gDNA extracted from HCT116 cells, which have both a WT and mutant (G13D) allele of *KRAS*. **f** DNA sequencing analysis of ORNi-RPA products. RPA and ORNi-RPA products in **e** were purified and sequenced using a forward primer. **g** Results of ORNi-RPA of gDNA possessing both a WT and mutant (G12D) allele of *KRAS*. **h** DNA sequencing analysis of ORNi-RPA products. RPA and ORNi-RPA products in **g** were purified and sequenced using a forward primer.

Fig. 3a). We introduced mutations in the *CDKN2A (p16)* gene in 293T cells and extracted gDNA for use in CRISPRi-RPA (Fig. 3h). Because the genome editing was highly efficient (Supplementary Fig. 7a and b), we mixed parental and genome-edited gDNA (4: 1) to mimic lowly efficient genome editing (Fig. 3i). We found that CRISPRi-RPA amplified only the edited *CDKN2A (p16)* sequences (Fig. 3i and j), suggesting that CRISPRi-RPA is useful in the evaluation of genome editing events.

**Blocking RPA using a DNA-binding protein**. Next, we examined DNA-binding proteins as sequence-specific blockers of RPA. LexA, a bacterial DNA-binding protein, strongly suppressed

amplification of a region of gDNA containing 8 copies of LexA-binding elements (LexA BE)[17] (Supplementary Fig. 8a–c), but not the irrelevant *KRAS* gene (Supplementary Fig. 8d), suggesting sequence-specific suppression of the RPA reactions. Thus, DNA-binding proteins can be used to suppress RPA reactions in a sequence-specific manner.

**Discrimination of CpG methylation status by blocking RPA using a methyl-CpG binding protein**. Lastly, we examined methyl-CpG binding domain protein 2 (MBD2)[18], a methylated CpG-binding protein often used for methylated DNA enrichment[19,20], as a potential epigenetic mark-specific blocker of RPA

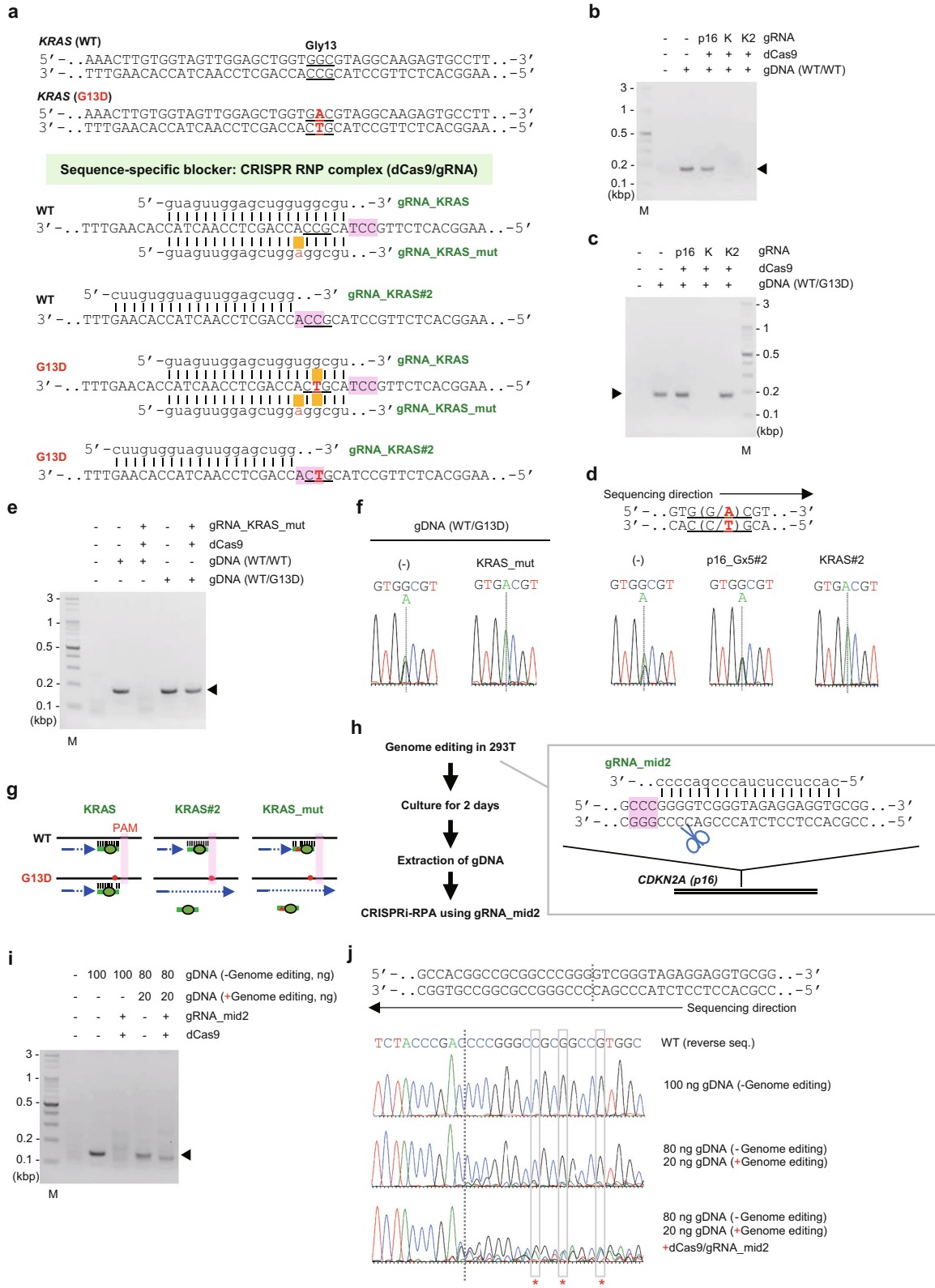

(Fig. 4a). To evaluate CpG methylation-specific suppression, we conducted RPA, in the presence of MBD2, of a CpG island in the *CDKN2A (p14ARF)* gene, which is highly methylated on one allele (Gx5 allele) but not the other (Gx4 allele) in HCT116 cells (Fig. 4b)[13]. As shown in Fig. 4c and d and Supplementary Fig. 9a and b, MBD2 suppressed the amplification of the *CDKN2A (p14ARF)* sequence (29 CpG sites) only from the methylated Gx5 allele of

HCT116 gDNA. Such methylated CpG-specific suppression was also observed for a CpG island (28 CpG sites) in the *CDKN2A (p16)* gene, which is similarly highly methylated on only one allele in HCT116 cells (Supplementary Fig. 10a–e)[13]. Thus, blocking RPA using MBD2 (named "MBD protein-interference RPA (MBDi-RPA)") can discriminate the methylation status of CpG islands. To confirm the direct involvement of CpG methylation in MBDi-RPA, we cloned

**Fig. 3 Blocking RPA using CRISPRi as a blocking agent. a** gRNAs and their target sequences in the human *KRAS* gene. **b**, **c**, **e** Results of CRISPR interference-RPA (CRISPRi-RPA) of gDNA extracted from 293T (**b**, **e**) or HCT116 cells (**c**, **e**). A schematic diagram of CRISPRi-RPA is shown in Fig. 1b and Supplementary Fig. 3a. Amplified *KRAS* is indicated by an arrowhead. M, molecular weight marker. **d**, **f** DNA sequencing analysis of CRISPRi-RPA products. CRISPRi-RPA products from **c** and **e** were purified and sequenced using a forward primer. **g** Mechanisms by which the discrimination of a single-nucleotide difference is achieved by CRISPRi-RPA. **h** A schematic diagram describing the evaluation of genome editing events by CRISPRi-RPA. **i** Results of CRISPRi-RPA. **j** Results of DNA sequencing analysis. RPA and CRISPRi-RPA products from **i** were purified and sequenced using a reverse primer. The CRISPR cleavage site is indicated by vertical dotted lines. The positions where intact *CDKN2A (p16)* nucleotides were not detected in the CRISPRi-RPA product are marked with asterisks.

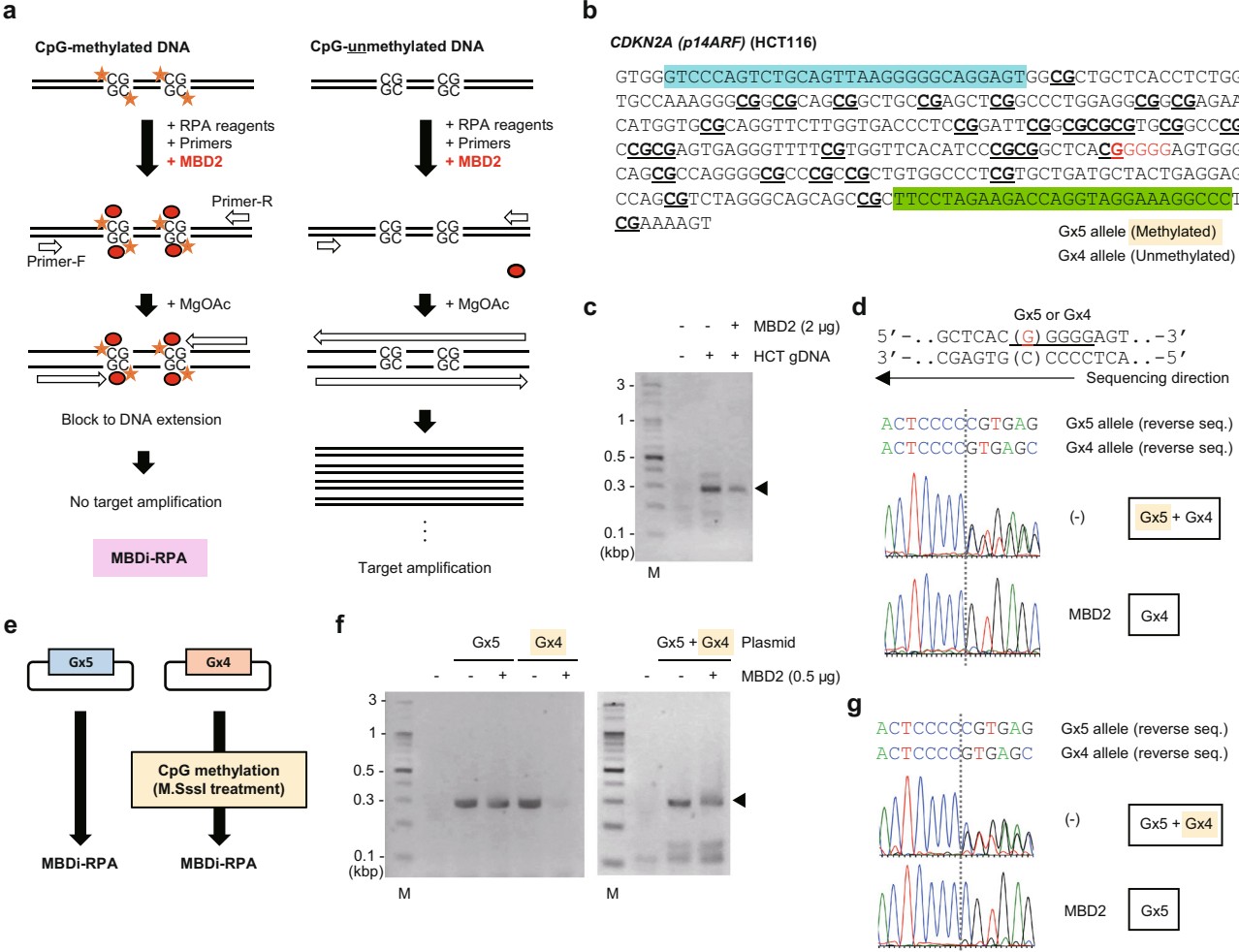

**Fig. 4 Blocking RPA to discriminate CpG-methylation status. a** A schematic diagram describing blocking RPA with MBD2 protein as a blocking agent (MBDi-RPA). MBD proteins bind to any methylated CpG sites in a target sequence, resulting in the suppression of amplification. MBDi-RPA can be used to discriminate CpG methylation status. **b** A target sequence in the human *CDKN2A (p14ARF)* gene in HCT116. Forward and reverse primer positions are highlighted in blue and green, respectively. **c** Results of MBDi-RPA. M, molecular weight marker. Amplified *CDKN2A (p14ARF)* is indicated by an arrowhead. **d** Results of DNA sequencing analysis. RPA or MBDi-RPA amplicons from **c** were sequenced using a reverse primer. **e** A schematic diagram describing MBDi-RPA of in vitro CpG-methylated plasmid DNA. The *CDKN2A (p14ARF)* sequences shown in **b** were cloned into plasmid vectors. The sequence corresponding to the Gx4 allele was subsequently methylated in vitro. **f** Results of MBDi-RPA from individual Gx4 and Gx5 plasmids or a plasmid mixture. **g** Results of DNA sequencing analysis. RPA and MBDi-RPA amplicons shown in **f** (plasmid mixture) were sequenced using a reverse primer.

both the Gx4 and Gx5 allele of the *CDKN2A (p14ARF)* CpG island into plasmid vectors and performed in vitro methylation (Fig. 4e). In this regard, we intentionally methylated CpG sites in the DNA sequence corresponding to the Gx4 allele, which is not CpG-methylated in HCT116 cells. MBDi-RPA specifically suppressed the amplification of methylated Gx4 DNA (Fig. 4f and g and Supplementary Fig. 11), demonstrating the direct involvement of CpG-methylation. These data show that MBDi-RPA is potentially useful for the discrimination of the CpG-methylation status of target DNA sequences.

## Discussion

By using ORNs, CRISPR complexes, and DNA-binding or methylated CpG-binding proteins as blockers, we have expanded the utility and flexibility of RPA to include the discrimination of nucleotide differences and epigenetic marks between cells, as well as detection of DNA-binding activities in proteins. We first showed that ORNi-RPA is useful to discriminate single-nucleotide differences (Fig. 2a–h). In this regard, we also tested a previously reported mode of blocking RPA using an ODN modified with 2'3'-dydeoxycitidine (2'3'ddC) at the 3'-end[6]

(Supplementary Fig. 12a). Because this ODN-mediated method interferes with the recruitment of a primer, its blocking mode is completely different from that of ORNi-RPA (Fig. 1b and Supplementary Fig. 12a). As shown in Supplementary Fig. 12b–e, we succeeded in amplifying the G13D *KRAS* sequence using a modified ODN, although amplification of the WT *KRAS* sequence was incompletely suppressed. Therefore, blocking RPA using a modified ODN can also be used to discriminate a single-nucleotide *KRAS* mutation without the additional heating/cooling step for ODN–template DNA hybridization. In general, the synthesis of an ODN is not expensive. However, the 2'3'ddC modification step is expensive, and therefore the total cost would be comparable with or higher than that of synthesis of an ORN. Because the modified ODN has to compete with a primer to block its annealing, its design may be more complicated than that of an ORN/primers for ORNi-RPA. Thus, from the viewpoint of design, ORNs might have advantages over 2'3'ddC-modified ODNs. By contrast, the stability of ODNs is generally better than that of ORNs, which is an advantage of ODNs. In this regard, because RPA reagents are provided as lyophilized forms, it would be interesting to include an ORN in lyophilized RPA reagents, which can be utilized as a diagnostic reagent to discriminate nucleotide mutations. ORNs would be stable under such conditions.

We showed that CRISPRi-RPA is also useful to discriminate single-nucleotide differences (Fig. 3a–g). CRISPRi-RPA is more expensive than ORNi-RPA because it requires the dCas9 protein and gRNA (crRNA/tracrRNA) to block DNA amplification. However, a gRNA would be much easier to design if a single-nucleotide difference disrupted a PAM sequence for a gRNA. Even if this is not the case, previous studies about tolerance of a single-nucleotide mismatch between a gRNA and target DNA[21,22] are useful to design an appropriate gRNA for discrimination of single-nucleotide differences by CRISPRi-RPA. In addition, it is possible to mutate a gRNA in order to intentionally form a tolerated nucleotide mismatch between the target DNA and gRNA sequence (e.g., gRNA_KRAS_mut). If there is another nucleotide difference in the target DNA (i.e., two nucleotide mismatches with the gRNA), such a DNA sequence would not be recognized by the gRNA. It would be interesting to compare the properties of ORNi-RPA and CRISPRi-RPA in more detail in the future.

CRISPR-mediated cleavage followed by PCR amplification can be used to discriminate nucleotide differences between samples[23]. Our approach, however, utilizes the CRISPRi system to block amplification by DNA polymerases in RPA, which can be conducted without specialized equipment and in a shorter period of time than PCR. It may be interesting to further investigate Cas9 as a block to target amplification during RPA at both the binding and cleavage steps. Furthermore, it would also be interesting to examine the CRISPR-associated complex for antiviral defense (Cascade) system as blocks to RPA, as these complexes arrest the elongation by DNA polymerases during DNA repair in adaptation events that lead to CRISPR immunity[24–27]. Moreover, it would be interesting to address whether the CRISPRi system can block DNA repair in eukaryote and prokaryote cells, especially in *Staphylococcus aureus* or *Bacillus subtilis*, from which the DNA polymerases for RPA are derived. Such studies may be useful for the development of CRISPRi-based elimination methods for specific pathogenic bacteria.

We show that DNA-binding proteins, such as LexA, successfully block RPA (Supplementary Fig. 8a–d). This blocking RPA assay can be used for the rapid evaluation of the DNA-binding activities of transcription factors (TFs) and other DNA-binding proteins. Several methods are used for the evaluation of DNA-binding activities, including the electrophoretic mobility shift assay (EMSA or gel shift assay)[28] and the enzyme-linked

immunosorbent assay (ELISA)[29]. However, these assays are laborious, while the RPA assay described in this study can be completed in under an hour. However, to achieve greater sensitivity and to expand the range of TFs assessed by this method, further system optimization would be required.

Bisulfite-conversion followed by methylation-specific PCR has been widely used for the evaluation of CpG methylation status[30]. MBDi-RPA does not require bisulfite conversion and can confirm the CpG-methylation status of CpG islands more rapidly. Notably, we found that two CpG-methylated sites are not sufficient to block RPA under the experimental conditions shown in Fig. 4f (Supplementary Fig. 13a–d), suggesting that MBD2 protein slows but does not arrest elongation of DNA polymerases in these circumstances. To expand the usefulness of MBDi-RPA, for single CpG-methylation or genome-wide analysis, for example, further optimization of experimental conditions such as the DNA polymerase activity is required. In addition to MBD2, other MBD family proteins[18], antibodies recognizing epigenetically modified DNA (e.g., 5-methylcytosine, 5-hydroxymethylcytosine, 5-carboxylcytosine, and 5-formylcytosine), and proteins binding to non-methylated CpG such as CXXC-Type Zinc Finger Protein 1[31] might be employed for blocking RPA to evaluate epigenetic marks on template DNA.

RPA can employ heat-sensitive molecules as blockers and amplifies DNA rapidly (~30 min), which is advantageous over PCR for many applications. However, the design of RPA primers is more difficult than PCR primers and while RPA can steadily amplify several hundred bp of DNA, it cannot amplify sequences longer than 1 kbp[3]. Indeed, we were sometimes unable to design optimal primer sets and non-specific products were amplified in some experiments (Supplementary Fig. 8b and c and 10b and d). If optimal primer sets were designed, quantitative blocking RPA using fluorescent dyes could be employed[2]. The techniques described here may also be applied to other isothermal amplification methods[32]. Alternatively, thermostable proteins and RNP complexes may be employed in PCR to discriminate nucleotide and epigenetic differences between samples. We believe that the data described here are useful in the development of such techniques for DNA analysis.

## Methods

**Oligonucleotides**. Primers were synthesized by Eurofins Genomics (Tokyo, Japan). ORNs, crRNA, and tracrRNA were chemically synthesized and purified by high-performance liquid chromatography (HPLC) (FASMAC, Kanagawa, Japan). An ODN modified with 2'3'ddC at the 3'-end was synthesized and purified by HPLC (Tsukuba Oligo Service, Ibaraki, Japan). Their sequences are listed in Supplementary Table 1.

**Cells and gDNA**. HCT116 was purchased from American Type Culture Collection (ATCC, Manassas, VA). 293T was from our lab stock. DT40#205-2 possessing eight copies of LexA BE was generated in our previous study[17]. gDNA was extracted from these cell lines by standard phenol/chloroform extraction. Alternatively, purified gDNA possessing the heterozygous G12D *KRAS* mutation (catalog number: HD272) was purchased from Horizon Discovery (Cambridge, UK).

**CRISPRi RNP complexes**. 3xFLAG-tagged *S. pyogenes* dCas9 protein (r3xFLAG-Sp-dCas9-D) was produced in our previous study[12]. To anneal crRNA and tracrRNA, 1 μl of crRNA (10 μM) and 1 μl of tracrRNA (10 μM) were annealed in 2 μl of nuclease-free water (total 4 μl) at 98 °C for 2 min and then cooled at room temperature. Thereafter, 0.4 μg of dCas9 protein and 0.8 μl of the crRNA/tracrRNA complex were mixed in nuclease-free water (total 10 μl) to form CRISPRi RNP complexes.

**LexA protein**. The 3xFLAG-tagged LexA protein was produced in our previous study[17].

**RPA reactions with gDNA**. RPA was performed using a TwistAmp™ Basic kit (TwistDx, Maidenhead, UK) as follows: a freeze-dried component was reconstituted with 29.5 μl of rehydration buffer and 2.5 μl of each primer (10 μM). A

13.6 µl aliquot was mixed with DNA (20 ng unless stated) to create a pre-reaction mixture. The required blocking agent (0.25–4 µM ORN, 1 µl of CRISPRi RNP complex, 20 or 40 ng of LexA, 0.25–2 µg of MBD2 from the EpiXplore™ Methylated DNA Enrichment Kit (Takara Bio, Shiga, Japan), or 0.5–2 µM 2'3'ddC-modified ODN) and nuclease-free water were added to the pre-reaction mixture to a final volume of 19 µl and incubated at 37 °C for 5 min (pre-incubation). Thereafter, 1 µl of MgOAc (280 mM) was added and the reaction was incubated at 37 °C for a further 30 min. RPA products were purified using a PCR/Gel DNA purification kit (Nippon Genetics, Tokyo, Japan), electrophoresed on 2% or 3% agarose gels, and sequenced if required. DNA gel images were acquired using AE-6905H Image Saver HR (ATTO, Tokyo, Japan) and the DigiDoc-It system (UVP, Cambridge, UK). DNA sequencing data were analyzed using Applied Biosystems Sequence Scanner Software v2.0 (ThermoFisher Scientific, Waltham, MA, USA).

**Genome editing**. 293T cells were cultured in Dulbecco's Modified Eagle's Medium (Wako, Tokyo, Japan) with 10% fetal bovine serum. For genome editing of the human *CDKN2A (p16)* locus, 293 T cells ($4 \times 10^5$) were transfected with a Cas9 expression plasmid (2 µg, Addgene #41815; a kind gift from Dr. George M. Church)[33] and a single gRNA expression plasmid targeting the human *CDKN2A (p16)* locus (gRNA_mid2, 2 µg)[13] using Lipofectamine 3000 (ThermoFisher Scientific). After 2 days, cells were collected for extraction of gDNA with a Quick-DNA™ Universal Kit (Zymo Research, Irvine, CA, USA).

**Plasmid construction and CpG methylation**. The *CDKN2A (p14ARF)* sequence shown in Fig. 4b was amplified by PCR and cloned into T-Vector pMD20 (Takara Bio). The plasmids were amplified in *E. coli* DH5α (Toyobo, Shiga, Japan) and purified. The plasmid including the DNA sequence corresponding to the Gx4 allele (1 µg) was CpG-methylated using CpG Methyltransferase M.SssI (New England Biolabs, Ipswich, MA, USA) and then purified using a PCR/Gel DNA purification kit (Nippon Genetics). Part of the human *EGFR* gene corresponding to the 2 CpG region (Supplementary Fig. 13b) was similarly amplified by PCR from 293T gDNA, cloned into T-vector pMD20, and CpG-methylated.

**MBDi-RPA reactions with plasmid DNA**. A pre-reaction mixture was prepared as described above with 1 pg of plasmid DNA. A total of 0.5 µg of MBD2 and nuclease-free water were added to the pre-reaction mixture to a final volume of 19 µl. After incubation at 37 °C for 10 min, 1 µl of MgOAc (280 mM) was added and the reaction was incubated at 37 °C for a further 10 min. RPA products were electrophoresed on 3% agarose gels and sequenced if required.

**Bisulfite-sequencing**. CpG-methylated or mock-treated plasmids were subjected to bisulfite treatment using the EZ DNA Methylation-Direct Kit (Zymo Research) followed by PCR using TaKaRa EpiTaq™ HS (for bisulfite-treated DNA) (Takara Bio). PCR cycles were as follows: 40 cycles of 98 °C for 10 s, 55 °C for 30 s, and 72 °C for 1 min. PCR products were cloned into T-Vector pMD20 (Takara Bio) and sequenced. Methylation status was analyzed by QUMA, a methylation analysis tool (http://quma.cdb.riken.jp/index_j.html) (Supplementary Fig. 11). Alternatively, PCR products were sequenced directly (Supplementary Fig. 13c).

**Statistics and reproducibility**. Experiments were performed at least in duplicate to verify the findings in Figs. 2–4 and Supplementary Figs. S2, S4, S5, S9, and S12.

**Reporting summary**. Further information on research design is available in the Nature Research Reporting Summary linked to this article.

## Data availability
All data and materials are included in this published article and Supplementary information.

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

## Acknowledgements

We thank Dr. George M. Church for providing the hCas9 plasmid (Addgene #41815). This work was supported by Hirosaki University Graduate School of Medicine (T.F. and H.F.), and the Karoji Memorial Fund for Medical Research in Hirosaki University (T.F.).

## Author contributions

T.F. and H.F. conceived the project and wrote the manuscript. T.F. designed the experiments. T.F. and S.N. performed the experiments.

## Competing interests

T.F. and H.F. have filed patent applications for ORNi-RPA, CRISPRi-RPA, and MBDi-RPA with details as follows: (1) Name: method for suppressing amplification of specific nucleic acid sequences; number: Japanese Patent Application Nos. 2014-176018 and 2015-165643; status: under review; specific aspect of manuscript covered in the patent application: the basic principle of ORNi-RPA described in this manuscript. (2) Name: method for detecting differences in the target nucleic acid region; number: Japanese Patent Application Nos. 2018-81752, 2019-555516, and PCT/JP2019/16843; status: registered; Japanese Patent No. 6653932; specific aspect of manuscript covered in the patent application: application of ORNi-RPA to detection of nucleotide differences.; (3) name: method for detecting target nucleic acid, a method for detecting molecule having the nucleic acid binding ability, and method for evaluating nucleic acid binding ability; number: Japanese Patent Application No. 2019-191409 and PCT/JP2020/39128; status: filed; specific aspect of manuscript covered in the patent application: application of CRISPRi-RPA and MBDi-RPA to detection of differences of nucleotides and epigenetic status. Epigeneron, Inc. owns the rights for the commercial use of ORNi-RPA, CRISPRi-RPA, and MBDi-RPA. T.F. and H.F. are co-founders of Epigeneron, Inc. and own stock in the company. H.F. is one of the directors of Epigeneron, Inc. The funders had no role in the design of the study; in the collection, analyses, or interpretation of data; in the writing of the manuscript, or in the decision to publish the results. The remaining author declares no competing interests.
