## [Transparent Peer Review File · Communications Biology]

Referee expertise:

Referee #1: Loop-Mediated Isothermal Amplification; Recombinases

Referee #2: Genomics, sequencing

Reviewers' comments:

Reviewer #1 (Remarks to the Author):

Authors have presented a study about blocking recombinase polymerase amplification for enabling the discrimination of DNA variants and epigenetic differences. The novelty is high because they propose the use of sequence-specific RNAs and proteins. I think that the idea is smart, useful and reliable for be published in a high-impact journal. The approach can be extended to many research areas. In fact, they have prepared an interesting patent. However, a scientific article requires that the conclusions must be supported in data. This manuscript lacks of some basic results for demonstrating the conclusions.

The developed method has been applied for discriminating single-nucleotide variant in KRAS gene. However, they have only tested wild-type and one variant (G13D). It is important to report the method performances in case of other variants are present.

There are some papers based on blocked RPA that the blocker is an oligonucleotide. Authors should design and perform experiments in order to compare their novel approaches to this option. Also, they have to deeply justify the performances because DNA oligonucleotides are more accessible, stable and cheaper for real applications than the proposed molecules.

It is important that the revised manuscript includes more experimental details. Any journal reader should be able to reproduce the experiment or to extend to new targets. Information about the ORN design respect to target sequence is missed. Experiments about the setup of the method are also needed, which are the concentration effect of blocker on assay performances? They must be in the main text and/or supplementary material.

I recommend that the description of previous studies be moved to introduction or discussion sections (example Lines 73-74, 100, 110 among other). Some sentences are ambiguous (lines 49-50). The direct application of RPA is incompatible to SYBR green dye, see the manufacturer instructions (Line 231).

Reviewer #2 (Remarks to the Author):

Fujita et al., describe a new method to detect specific DNA sequences based on elongation-blocked RPA (Recombination Polymerase Amplification). The three classes of blockers described in the manuscript are: RNA oligonucleotides, heat-sensitive protein-RNA complexes, and heat-sensitive DNA-binding proteins. Although the examples described in the manuscript are limited, the technique can be further expanded and applied to numerous fields such as molecular diagnostics and potentially compatible with point-of-care testing.

The experimental design, results, and interpretation are presented clearly.

Comments on minor improvements are reported below.

The manuscript has the potential of being of high impact since the methodology can be used in several applications, ranging from SNPs detection, DNA-binding activity evaluation, and epigenetic applications. As the authors state in the manuscript, the method can (and should) be optimized for certain applications.

Comments:

1. Figure 1E/F: It would be ideal that the authors show that they are able to suppress the amplification of the mutated allele (G13D) with ORN. The Sanger sequencing results should show only the G nucleotide.
2. Figure 1F: since the KRAS sequences (wt and G13D) in figure1B are the reverse of the ones reported in 1F, please add (in 1F) that A represents the mutated allele (G13D), while G represents the wt allele for easier comprehension.
3. Figure 2J: it would be ideal to show a trace of the Sanger sequencing results of the genome-edited gDNA for a better comparison
4. Supplementary Figure S6: it would be interesting if the authors could apply the RPA to a DNA with a reduced number of LexA-binding elements in order to determine what is the minimum number that can block the amplification. Although each DNA-binding protein's affinity for DNA is different, it would be a good proof-of-principle experiment.
5. Figure 3D and 3G: I think the figure is misleading. In the PCR reaction, both Gx4 and Gx5 alleles are present, but as seen in the Sanger results, only one of the two is amplified when MBD2 is present. I think it would be ideal to specify that the boxed Gx4/Gx5 on the right of the panels are the detected sequences by the Sanger approach.

Responses to Reviewer Comments

We thank the editor and reviewers for their helpful comments to improve the manuscript again. We corrected the manuscript according to their suggestions. Revised wordings are shown in red in the revised document. Please find below our responses to specific issues raised by the reviewers.

Reviewer #1:

Authors have presented a study about blocking recombinase polymerase amplification for enabling the discrimination of DNA variants and epigenetic differences. The novelty is high because they propose the use of sequence-specific RNAs and proteins. I think that the idea is smart, useful and reliable for be published in a high-impact journal. The approach can be extended to many research areas. In fact, they have prepared an interesting patent.

We thank the reviewer for appreciating the utility of blocking RPA methods that we established in this study.

1. However, a scientific article requires that the conclusions must be supported in data. This manuscript lacks of some basic results for demonstrating the conclusions. The developed method has been applied for discriminating single-nucleotide variant in *KRAS* gene. However, they have only tested wild-type and one variant (G13D). It is important to report the method performances in case of other variants are present.

We thank the reviewer for the insightful comments. According to the reviewer's comments, we performed ORNi-RPA and CRISPRi-RPA using genomic DNA that possesses another well-known *KRAS* mutation, G12D. We found that both ORNi-RPA and CRISPRi-RPA suppress amplification of the target (WT) sequence, resulting in detection of the G12D *KRAS* (new Figure 2G and H, new Supplementary Figure S2, and new Supplementary Figure S4). Thus, those data strongly support these blocking RPA methods can discriminate other nucleotide mutations.

2. There are some papers based on blocked RPA that the blocker is an oligonucleotide. Authors should design and perform experiments in order to compare their novel approaches to this option. Also, they have to deeply justify the performances because DNA oligonucleotides are more accessible, stable and cheaper for real applications than the proposed molecules.

We also thank the reviewer for the insightful comments. According to the reviewer's comments, we tested blocking RPA using an oligodeoxyribonucleotide (ODN) modified with 2'3'-dydeoxycytidine (2'3'ddC) at the 3'-end, which is necessary to avoid DNA extension. We first designed an ODN targeting a *KRAS* mutation, G13D, according to the previous report (reference #6). We next performed this blocking RPA to detect the G13D *KRAS*. We could detect the G13D *KRAS* although the suppression of the WT *KRAS* was incomplete. Therefore, blocking RPA using a 3'-modified ODN can also be used to discriminate a single-nucleotide *KRAS* mutation although optimization may be required. We showed those results in new Supplementary Figure S12 and the Discussion section because the section would be more suitable to discuss properties of this method and ORNi-RPA.

As shown in new Supplementary Figure S12A, the blocking mode of ODN is different from that of ORNi-RPA. Therefore, it would be difficult to compare the utility of both methods in detail. However, we described properties of both methods as much as possible in the Discussion section. For example, in the view of synthesis cost of the 3'-modified ODN, it is not necessarily advantageous over that of an ORN because the 2'3'ddC modification, which was concluded as a better modification in the previous report (reference #6), is expensive. In addition, because a limited company can provide the 2'3'ddC modification (i.e., this modification may be specific), an ODN with the 2'3'ddC modification is not necessarily more accessible than an ORN. Moreover, because a modified ODN has to compete with a primer to block its annealing, their design may be more complicated than that of an ORN/primers for ORNi-RPA. However, stability of ODNs is generally better than that of ORNs, which is an advantage of ODNs. In this regard, because RPA reagents are provided as lyophilized

forms, it might be interesting that an ORN is also involved in lyophilized RPA reagents as a diagnostic reagent to discriminate nucleotide mutations. ORNs would be also stable under such a condition.

3. It is important that the revised manuscript includes more experimental details. Any journal reader should be able to reproduce the experiment or to extend to new targets. Information about the ORN design respect to target sequence is missed. Experiments about the setup of the method are also needed, which are the concentration effect of blocker on assay performances? They must be in the main text and/or supplementary material.

According to the reviewer's comments, we suggested potential a step-by-step procedure for ORNi-RPA to detect a single-nucleotide difference in new Supplementary Figure S2E and F, based on the results of ORNi-RPA. We also added description that titration of an ORN would be beneficial to optimize the assay systems (Lines 114-115). We believe that those procedures are useful to target new sequences by ORNi-RPA. Other technical information was moved from the Supplementary Text and is now shown in detail in the Methods section in the main text.

4. I recommend that the description of previous studies be moved to introduction or discussion sections (example Lines 73-74, 100, 110 among other).

According to the reviewer's comments, we moved related sentences to the last paragraph of the Introduction section.

5. Some sentences are ambiguous (lines 49-50).

We revised the sentences. In addition, we added other sentences to make clearer what we mean (lines 47-56).

6. The direct application of RPA is incompatible to SYBR green dye, see the manufacturer instructions (Line 231).

We thank the reviewer for this information. We deleted “SYBR Green I” to avoid readers' confusion although the first paper on RPA (reference #2) showed real-time RPA using SYBR Green I.

Reviewer #2:

Fujita et al., describe a new method to detect specific DNA sequences based on elongation-blocked RPA (Recombination Polymerase Amplification). The three classes of blockers described in the manuscript are: RNA oligonucleotides, heat-sensitive protein-RNA complexes, and heat-sensitive DNA-binding proteins. Although the examples described in the manuscript are limited, the technique can be further expanded and applied to numerous fields such as molecular diagnostics and potentially compatible with point-of-care testing. The experimental design, results, and interpretation are presented clearly. Comments on minor improvements are reported below. The manuscript has the potential of being of high impact since the methodology can be used in several applications, ranging from SNPs detection, DNA-binding activity evaluation, and epigenetic applications. As the authors state in the manuscript, the method can (and should) be optimized for certain applications.

We thank the reviewer for appreciating the utility of the blocking RPA methods that we established in this study.

1. Figure 1E/F: It would be ideal that the authors show that they are able to suppress the amplification of the mutated allele (G13D) with ORN. The Sanger sequencing results should show only the G nucleotide.

We thank the reviewer for the helpful comment. In this study, we applied ORNi-RPA to detection of a *KRAS* mutation sequence, G13D, because it would be useful for diagnosis

of cancers accommodating this mutation in future. In this regard, detection of the WT *KRAS* sequence (i.e., suppression of the G13D *KRAS*) might not be attractive in clinical settings. However, because examples of ORNi-RPA were limited in the original manuscript, we speculate that the reviewer might suggest suppressing the mutation G13D as another example. Therefore, in the revised manuscript, we demonstrated another example on detection of the G12D *KRAS* mutation using other ORNs, which would be interesting for readers. Please see the response to the comment #1 of the reviewer #1 for more detail.

2. Figure 1F: since the *KRAS* sequences (wt and G13D) in figure1B are the reverse of the ones reported in 1F, please add (in 1F) that A represents the mutated allele (G13D), while G represents the wt allele for easier comprehension.

According to the reviewer's comment, we added sequence information in new Figure 2F (previous Figure 1F) and other related figures to clearly distinguish the mutated and WT alleles, which will avoid readers' confusion.

3. Figure 2J: it would be ideal to show a trace of the Sanger sequencing results of the genome-edited gDNA for a better comparison.

We also thank the reviewer for the helpful comment. In general, indels of genome editing should be various so that many types of edited sequences would be included in the pool of genome-edited gDNA. Therefore, it would be difficult to trace all the mutated sequences because it is necessary to clone them for sequencing. We believe that the most important message of this figure is to show that the WT sequence is not detectable. In this context, we think that the current form of this figure (now Figure 3J) would deliver this message. Therefore, we keep this figure as it is.

4. Supplementary Figure S6: it would be interesting if the authors could apply the RPA to a DNA with a reduced number of LexA-binding elements in order to determine what

is the minimum number that can block the amplification. Although each DNA-binding protein's affinity for DNA is different, it would be a good proof-of-principle experiment.

We thank the reviewer for the insightful comments. We are also interested in the minimum number of LexA-binding elements to block DNA amplification. In this regard, we used the LexA protein just as an example to examine whether a protein can be used as a blocker for blocking RPA. In this context, the data shown in the Supplementary Figure S6 (now Supplementary Figure S8) would demonstrate this feasibility. In fact, as shown in Figure 3 (now Figure 4), the MBD2 protein can also be used as a protein blocker.

We aim to apply this protein-based blocking RPA system to analysis of binding activities of various DNA-binding proteins including transcription factors. This approach should be much easier than other analytical methods, such as gel shift assay, because it only requires addition of test DNA-binding factors into RPA reaction mixture. To this end, the use of multiple copies of protein binding elements may achieve higher sensitivity than that of a single copy. The data shown in the Supplementary Figure S6 (now Supplementary Figure S8) would demonstrate the feasibility of such an application.

Thus, although examination of the minimum number of LexA-binding elements to block DNA amplification would be an interesting issue, it is out of scope of this study and would be addressed in the future.

5. Figure 3D and 3G: I think the figure is misleading. In the PCR reaction, both Gx4 and Gx5 alleles are present, but as seen in the Sanger results, only one of the two is amplified when MBD2 is present. I think it would be ideal to specify that the boxed Gx4/Gx5 on the right of the panels are the detected sequences by the Sanger approach.

According to the reviewer's comment, we added sequence information in new Figure 4D and G (previous Figure 3D and G) to clearly distinguish Gx4/Gx5 (Cx4/Cx5 for complementary sequences), which will avoid readers' confusion.

REVIEWERS' COMMENTS:

The authors addressed satisfactorily all the reviewer's comments.